# Prognostic Biomarkers in Endometrial Cancer: A Systematic Review and Meta-Analysis

**DOI:** 10.3390/jcm9061900

**Published:** 2020-06-17

**Authors:** Eva Coll-de la Rubia, Elena Martinez-Garcia, Gunnar Dittmar, Antonio Gil-Moreno, Silvia Cabrera, Eva Colas

**Affiliations:** 1Biomedical Research Group in Gynecology, Vall Hebron Institute of Research, Universitat Autònoma de Barcelona, CIBERONC, 08035 Barcelona, Spain; evacolldelarubia@gmail.com; 2Quantitative Biology Unit, Luxembourg Institute of Health, L-1445 Strassen, Luxembourg; elena.martinezgarcia@lih.lu (E.M.-G.); gunnar.dittmar@lih.lu (G.D.); 3Gynecological Department, Vall Hebron University Hospital, CIBERONC, 08035 Barcelona, Spain

**Keywords:** endometrial cancer, protein biomarker, prognostic, prognosis, risk assessment, ESMO-ESGO-ESTRO risk classification, TCGA, endometrial adenocarcinoma, uterine cancer, recurrence

## Abstract

Endometrial cancer (EC) is the sixth most common cancer in women worldwide and its mortality is directly associated with the presence of poor prognostic factors driving tumor recurrence. Stratification systems are based on few molecular, and mostly clinical and pathological parameters, but these systems remain inaccurate. Therefore, identifying prognostic EC biomarkers is crucial for improving risk assessment pre- and postoperatively and to guide treatment decisions. This systematic review gathers all protein biomarkers associated with clinical prognostic factors of EC, recurrence and survival. Relevant studies were identified by searching the PubMed database from 1991 to February 2020. A total number of 398 studies matched our criteria, which compiled 255 proteins associated with the prognosis of EC. MUC16, ESR1, PGR, TP53, WFDC2, MKI67, ERBB2, L1CAM, CDH1, PTEN and MMR proteins are the most validated biomarkers. On the basis of our meta-analysis ESR1, TP53 and WFDC2 showed potential usefulness for predicting overall survival in EC. Limitations of the published studies in terms of appropriate study design, lack of high-throughput measurements, and statistical deficiencies are highlighted, and new approaches and perspectives for the identification and validation of clinically valuable EC prognostic biomarkers are discussed.

## 1. Introduction

### 1.1. Endometrial Cancer

Endometrial cancer (EC) is the sixth most common cancer in women worldwide and the most common tumor of the female genital tract. A total number of 382,069 new estimated cases and 89,929 estimated deaths were reported for 2018 [1]. EC incidences have been increasing in the last years as a consequence of the populations increasing life expectancy and a higher overall prevalence of obesity and metabolic syndromes. Moreover, unlike many other malignancies, EC mortality has also been increasing [2]. The number of new cases and deaths is expected to increase by 20.3% and 17.4% by 2025, respectively [1]. Mortality of EC patients is directly associated to the presence of poor prognostic factors, which drive tumor recurrence.

### 1.2. Current Diagnosis and Preoperative Risk Stratification of Ec Patients

To date, screening tests of EC do not exist. Thus, current diagnosis relies on the presentation of symptoms, with abnormal uterine bleeding (AUB) being the most common one, since it is present in 90% of EC patients. However, this symptom is common to other diseases and only 10–15% of women with AUB will develop EC [3,4]. In order to rule out EC from other benign diseases, patients undergo a multi-step process of diagnosis including a gynecological examination, transvaginal ultrasonography (TVUS) and a pathological examination of an endometrial biopsy. This biopsy can be obtained by different procedures, i.e., aspiration (office-based pipelle biopsy), dilatation and curettage (D&C), or hysteroscopy, depending on the clinician’s choice [5]. Pathologists use these endometrial biopsies to give a final diagnosis of EC and prognostic information. Once EC is diagnosed, blood and imaging tests will also provide additional prognostic information of the tumor to finally reach a presurgical stratification, which will ultimately guide the surgical treatment.

A prognostic factor is a patient or disease characteristic that provides information about the likely outcome of the disease, independent of therapy, at the time of diagnosis [6]. The most important prognostic factors of EC patients include tumor grade, histological subtype, depth of myometrial invasion, cervical involvement, tumor size, lymphovascular space invasion (LVSI) and lymph node status [5].

Information about tumor grade and histological subtype is obtained from the histological examination of endometrial biopsies. Traditionally, EC has been classified into two different subtypes based on clinical, pathological and molecular features. The most common is the Type I or endometrioid (EEC) subtype, which mostly includes the endometrioid histology and is associated with a good prognosis; while Type II or non-endometrioid (NEEC), includes different minor histologies such as serous (~10%), clear cell (~3%), mixed cell adenocarcinoma and other rare types, and is associated with a poor prognosis [7] (Figure 1A). Although this dualistic classification is broadly used in the current clinical practice for preoperative assessment and surgical planning [8], its prognostic value remains limited. Using this classification model, approximately 20% of EEC cases relapse, whereas 50% of NEEC do not [7]. Besides the binary classification, 15–20% of EEC tumors are classified as high-grade lesions that do not fit in this model (Figure 1B).

Imaging techniques represent a cornerstone in the preoperative evaluation of patients with EC. Myometrial invasion, tumor size, and cervical invasion are properly assessed using imaging techniques such as TVUS, or magnetic resonance imaging (MRI). Lymph node status can be studied by computed tomography (CT), MRI or positron emission tomography (PET-CT), and those techniques seem to have similar accuracies. Specificity is high but the sensitivities are moderate to low [9]. In fact, imaging cannot replace lymph-node dissection for staging purposes since the detection rate for metastatic lesions of 4mm or less is only 12% [10]. Another important prognostic factor is the LVSI since the presence of tumor cells within vascular spaces is considered an early step in the metastatic process and consequently a strong predictor of nodal metastasis, recurrence and cancer-specific death [11]. However, it cannot be assessed using imaging techniques and usually not observed in preoperative biopsies. Thus, the decision to perform lymphadenectomy can be made based on intra-operative frozen sections or preoperative risk assessment based on histology and imaging tests, where the most important criteria are myometrial invasion and tumor grading. In this regard, sentinel lymph node mapping has emerged as a new strategy to know lymph node status while sparing many lymph node dissections [12].

### 1.3. Risk Stratification Systems

Once prognostic factors are identified in the resected tumor tissue, i.e., after surgical treatment, patients are classified according to the risk stratification system defined by the three major consortiums in EC, i.e., the ESMO-ESTRO-ESGO consensus [13]. As seen in Figure 2, this system purely uses the clinical, molecular and pathological information to define groups of patients from low to metastatic, measuring the risk of recurrence to classify, treat and predict the outcome of EC patients. Although this classification has the highest power of discrimination for stratifying the risk of recurrence in patients with EC, still 9% of patients at low risk recur while 60% of patients at high risk will not recur [14]. Therefore, it is evident that an improved version of the current stratification system needs to be developed, probably by the integration of additional molecular biomarkers.

In 2013, The Cancer Genome Atlas (TCGA) Research Network proposed a novel classification system exclusively based on a profound molecular characterization of the tumors [15]. According to this, EC is classified into four different subgroups: POLE ultramutated, microsatellite stability instable (MSI) hypermutated, copy-number low (microsatellite stable, MSS), and copy-number high (serous-like). The POLE ultramutated group is a small group of all grades EEC and barely a few serous EC [16], and is characterized for its excellent prognosis. The MSI hypermutated and copy-number low (MSS) groups are composed of endometrioid tumor histology and they have an intermediate prognosis. They are distinguished because MSI hypermutated subgroup includes most high-grade endometrioid cancers displaying genomic instability, while the copy-number low (MSS) subgroup has a lower rate of somatic copy number alterations. Tumors classified as copy-number high (serous-like) present the worst prognosis. They include serous EC and high-grade EEC with the highest load of somatic copy number alterations (Figure 3).

In order to translate this system into clinical practice, the ProMisE decision-tree and the TransPORTEC classifications emerged, mainly based on the assessment of three surrogate biomarker tests: (i) POLE sequencing; (ii) immunohistochemistry of mismatch repair proteins (MMR-IHQ); (iii) immunohistochemistry of the p53 protein [17,18,19]. These surrogate biomarkers categorize most ECs according to their molecular classification, but the method is limited by (i) systematic evaluation of the pathogenicity of POLE mutations is required [20]; (ii) p53 immunohistochemistry does not perfectly correlate with TP53 copy-number alterations; (iii) tumors harboring more than one classifying genomic aberration are difficult to classify; (iv) the algorithms do not include the evaluation of the significant heterogeneity seen in the copy-number-low group [21]. Despite these limitations, the scientific and clinical community supports incorporating the TCGA molecular classification in daily clinical practice for classifying EC patients. Indeed, there exists a high correlation of the TCGA classification between preoperative biopsies and the resected tumor tissue, thus, allowing us to incorporate this information preoperatively to guide both surgical and adjuvant treatment [22].

The TCGA molecular classification promises to improve the current ESMO-ESGO-ESTRO risk stratification system since it provides additional prognostic information. Indeed, studies performed in large cohorts of patients, notably by the TCGA (additional cohort), the Vancouver and the PORTEC groups [18,19,23,24,25], validated its prognostic relevance and pointed out specific subsets of patients who will benefit from this classification system (Figure 4). In particular, it has been reported that 7% of patients diagnosed with a good prognosis cancer (EEC-Grade 1) but with a copy-number high molecular diagnosis will now be stratified in a poorer prognosis group [15]. In contrast, all patients with POLE-hypermutated tumors (6–13% of all ECs) will now be considered good prognosis tumors, independently of the state of other prognostic factors (e.g., histological grade or FIGO stage).

## 2. Aim of the Review

In the age of personalized medicine, the detailed classification of patient subgroups is imperative before and after surgical treatment. For the EC, this translates into the improvement of stratification tools, including pathologic parameters, imaging techniques and molecular markers. The TCGA molecular classification offers a basis for such an integrative approach.

In this review, we systematically reviewed the existing literature compiling an overview of the numerous proteins which are EC prognostic factors associated or are directly related to recurrence and survival. Among those, we highlight the proteins with an increased potential to become prognostic biomarkers in the clinical setting after prospective validation. Finally, we discuss possible improvements and new approaches not yet applied to EC biomarker research that could accelerate the identification of clinically relevant biomarkers.

## 3. Materials and Methods

### 3.1. Search Strategy

Literature searches were performed in MEDLINE from 1991 to February 2020 using the terms “endometrial cancer” or “endometrial neoplasms” or “endometrial carcinoma”, “biomarkers” or “markers”, and “prognosis or prognostic” or “recurrence” or “survival”.

### 3.2. Screening

Duplicate hits were discarded. Unrelated studies were excluded through careful browsing of the title and/or abstract of each publication. Articles where only the abstract was available were also rejected.

### 3.3. Inclusion and Exclusion Criteria

The inclusion criteria were (1) studies including endometrial cancer with an epithelial origin; (2) biomarker studies performed at protein level; (3) prognostic biomarker studies, i.e., studies that identify or validate biomarkers that are associated to EC risk factors, recurrence or survival; (4) studies performed on any biological human sample, but not on cultured cells or animal models; (5) studies based on the expression of biomarkers. Exclusion criteria were articles (1) not written in English; (2) based on the characterization of one specific EC subtype; (3) based on response-to-treatment biomarkers; (4) articles performed using less than 10 samples in total; (5) reviews, meta-analyses, opinion articles or case report studies.

### 3.4. Data Extraction

All selected articles were reviewed and data were compiled in a comprehensive database which contained: general information (name of the first author, country, journal, year of publication); number of patients and analytical technique used; association of the described biomarkers with different prognostic factors (histological type, histological grade, FIGO stage, myometrial invasion, lymph node status, LVSI, cervical invasion, metastasis, TCGA molecular classification, recurrence, risk, overall survival (OS), disease-free survival (DFS), disease-specific survival (DSS), progression-free survival (PFS), and recurrence-free survival (RFS)); statistical information of the identified biomarkers (e.g., *p*-value, adjusted *p*-value, fold-change, area under the receiver operating characteristic curve (AUC), etc.).

### 3.5. Quality Assessment

The guidelines from Reporting Recommendations for Tumor Marker Prognostic Studies (REMARK) [27,28] were used to evaluate the quality of studies that were eligible.

### 3.6. Functional Enrichment Analysis

To investigate the potential functions of the most studied proteins regarding EC prognosis, we performed Gene Ontology (GO) and Kyoto Encyclopedia of Genes and Genomes (KEGG) pathway analysis using Database for Annotation, Visualization and Integrated Discovery (DAVID) (https://david.ncifcrf.gov/home.jsp). The GO terms refer to biological processes (BP) [29]. KEGG was used to identify the most deregulated EC pathways [30].

### 3.7. Statistical Analysis

A meta-analysis on OS was performed for the five most studied biomarkers. Only studies providing an estimate of the hazard ratio (HR) and the associated 95% CI for the parameter here considered were included. Since not all studies provided the same data regarding the estimation of HR, we focused the meta-analysis mainly on the unadjusted or ‘univariate’ estimates of HR. However, we used adjusted or ‘multivariate’ estimates for those articles not providing the unadjusted. HR and confidence interval (CI) were log-transformed. Pooled estimates of the HR (overall-effect model), and statistics *I^2^* and tau-squared were computed following the guidelines of *Doing Meta-Analysis in R* [31]. Analysis and forest plots were created using the ‘meta’ package (Schwarzer, 2007) of the R software (R Core Team, 2019).

### 3.8. Analyses of TCGA Data

Data from The Cancer Genome Atlas (TCGA) cohort of uterine corpus endometrial carcinoma, published in Nature [15], was obtained from https://tcpaportal.org/tcpa/download.html (L4) and clinical data were retrieved using cBioPortal. Protein expression levels were plotted using data from 200 patients and R software (R Core Team, 2019).

### 3.9. Analyses of CPTAC Data

Data used in this publication were generated by the Clinical Proteomic Tumor Analysis Consortium (NCI/NIH). Thermo RAW files and clinical data from the Clinical Proteomic Tumor Analysis Consortium (CPTAC) Uterine Corpus Endometrial Carcinoma (UCEC) Discovery study published in Cell [32] were retrieved using https://cptac-data-portal.georgetown.edu. MaxQuant software package version 1.6.7.0 [33] and the human database from Uniprot [34] were used to perform the protein and peptide identification and quantification. Protein expression levels from 100 patients were plotted using the proteinGroups.txt file by using R software (R Core Team, 2019).

## 4. Results

### 4.1. Data Summary

Our search retrieved 2507 hits in the initial PubMed Search, that were reduced to 1557 after the first screening step. Of those, 398 met our criteria and were included in this review (Figure 5A). Biomarker research on prognostic biomarkers in EC has increased over time and the global distribution points to Asia (43%) and Europe (41%) as the main contributors. At the country level, the leading countries are Japan, China, the United States of America, Turkey and Norway (Figure 5B).

From the 398 reviewed studies, a total of 255 protein biomarkers were identified as potential prognostic biomarkers, defined as proteins that are associated with one or more of the known clinical prognostic factors in EC, recurrence or survival. Remarkably, only 6% of articles have categorized the recruited patients and/or analyzed their results based on the TCGA classification from 2013 to date (Figure 5C). From the 255 protein biomarkers compiled in this review, only 21% were validated by using either an independent technique, an independent cohort, or in an independent study. Curiously, 60% of the studies were based on the study of a single protein (Figure 5D). Regarding the clinical sample used, 79% of the studies were performed in tissue specimens, followed by 16% of studies that used serum samples. Other sources were plasma, imprint smears, peritoneal cytology or uterine aspirates. Additionally, six studies were performed in tissue and validated in serum samples and five articles did it viceversa (Figure 5E).

### 4.2. Prognostic Protein Biomarkers in EC

As shown in Figure 6, the majority of biomarkers identified in this systematic review were associated with histological grade, FIGO stage and OS, with more than 100 biomarkers described for each of these parameters. Other biomarkers were associated with lymph node status, histological type, myometrial invasion, LVSI, DFS, recurrence, DSS, PFS, risk, RFS, metastasis, cervical invasion and the TCGA subgroups (Figure 6). The vast majority of biomarkers are related to more than one of the above-mentioned parameters, indicating that they provide relevant prognostic information but are not specifically linked to one feature in particular. In fact, those that were associated with a specific parameter (in bold in Figure 6) generally corresponded to those biomarkers that have been scarcely studied. Thus, further research needs to be performed to understand whether they are truly significant as prognostic factors and specific of that parameter in particular or might be also related to other parameters.

Most of the candidate prognostic biomarkers are involved in common biological processes, such as cellular processes, biological regulation, metabolic processes, response to a stimulus, cellular component organization or biogenesis and signaling and also, proteins that are part of the basic structural and functional units of the cell. In order to become promising biomarkers, the potential of any specific protein should be validated in different cohorts, and if possible, by independent groups, and in prospective studies. According to our findings, 11 proteins have been extensively studied, i.e., in more than five independent studies (Figure 7A). Importantly, all these proteins have also been described as diagnostic biomarkers and are the main drivers of the oncogenic pathways related to EC (Figure 7B,C) [35,36].

*ERBB2* codes for a protein tyrosine kinase. In the nucleus of the cell it is involved in transcription regulation and it enhances protein synthesis and cell growth. ERBB2 has been reported as an amplified oncogene in cancer and its protein overexpression has been associated with high grade and high-stage, NEEC histologies, the degree of tumor progression and the outcome and survival of the patients [37]. Additionally, it has demonstrated to be a potential therapeutic target in serous EC tumors overexpressing ERBB2 [38]. *CDH1* codes for e-cadherin, a calcium-dependent cell adhesion protein involved in mechanisms regulating cell-cell adhesions, mobility and proliferation. In EC, it has been associated with the epithelial-mesenchymal transition (EMT) and has demonstrated to have a role in the progression of EC [39]. Its loss has been associated with worse prognostic factors and poorer survival [40]. *PTEN* is the most frequently mutated gene in EC. *PTEN* is a tumor suppressor and it plays a role in cell cycle progression and cell survival by modulating the AKT-mTOR signaling pathway. In EC, the loss of function of *PTEN* has been postulated as an early event in carcinogenesis and correlated to a good prognosis [41]. *TP53* is a tumor suppressor protein involved in cell cycle regulation, growth arrest and apoptosis, and it is involved in activating oxidative stress-induced necrosis. As one of the guardians of the genome, it is one of the most mutated genes in a wide range of cancers. The measurement of p53 by immunohistochemistry is broadly used in EC to classify tumor subtypes since p53 overexpression is linked to high-grade endometrioid and serous subtypes [42]. MMR proteins are components of the post-replicative DNA mismatch repair system involved in DNA repair. Defects in these proteins resulted in the definition of a new subgroup of the TCGA molecular classification: phenotype MSI, mainly caused by methylation of the MLH1 promoter and associated to type I EC [15]. The estrogen and progesterone receptors (ESR1 and PGR) are involved in the regulation of eukaryotic gene expression and affect cellular proliferation and differentiation. In EC, the positivity of these receptors has been associated with type I and a good prognosis [43]. The loss of both proteins was reported to independently predict lymph node metastasis with a specificity of 0.84 [44]. *MKI67* is a proliferation marker with a role in maintaining individual mitotic chromosomes dispersed in the cytoplasm following nuclear envelope disassembly. Its labeling index is low in low-grade squamous areas of EC [43]. Its expression has been related to the early stages of the disease, as well as worse OS [45]. Besides its potential to predict lymph node metastasis with an AUC of 0.604 [46]. The neural cell adhesion molecule L1 (L1CAM) is involved in the dynamics of cell adhesion. It has been described as critical in EC to promote the EMT and predictive of worse outcomes among EC, including tumors diagnosed at an early stage. L1CAM was described as the best-ever published prognostic factor able to greatly predict recurrence (sensitivity of 0.74; specificity of 0.91) and death (sensitivity of 0.77; specificity of 0.89) [47]. Finally, the glycoproteins *MUC16* and *WFDC2* are two proteins widely evaluated, since their high expression demonstrated either diagnostic and prognostic value in gynecological tumors. Specifically, MUC16, an epithelial ovarian carcinoma antigen, is already used in clinics as a tool to diagnose ovarian cancer and it has been shown to predict lymph node metastasis with 0.78 sensitivity and specificity in EC [48]. Remarkably, in EC, serum WFDC2 reported a significantly higher pooled sensitivity (0.71) in comparison to MUC16 (0.35) and has been demonstrated to be a better diagnostic tool [49]. WFDC2 has also been demonstrated to be a more sensible predictor of lymph node metastasis (sensitivity of 0.82) in relation to MUC16 (0.72) [50] and better in predicting myometrial invasion (AUC of 0.76) than MUC16 (AUC = 0.65) [12].

All the 11 proteins have been studied by immunohistochemistry in primary tissue specimens. Notably, MKI67 and PTEN were also validated in tissue samples from imprint smears [51,52,53], as well as CDH1, which was analyzed in uterine aspirates using mass spectrometry-based approaches [54]. MUC16 and WFDC2 have been extensively studied in serum samples by antibody-based techniques such as ELISA or chemiluminescence techniques [50,55,56,57] and L1CAM was also validated in serum but just in one study [58].

### 4.3. Meta-Analysis of the Top-5 Most Studied Biomarkers

Results of the meta-analysis on OS of the five most studied proteins are shown in Figure 8. The number of investigations meeting our criteria for the estimation of HR was low, ranging from five to seven articles per protein. Substantial heterogeneity in the HRs across studies was observed for MUC16 and PGR, where the point estimates of the pooled HR and the 95% CI from the fixed and random effects model were wide. On the contrary, in ESR1, TP53, and WFDC2 the point estimates and the error margins were similar. Focusing on the data available, there is not enough evidence to affirm that MUC16 and PGR are useful EC prognostic biomarkers. However, articles studying ESR1, TP53 and WFDC2 point out these biomarkers as promising to be prognosticators of OS. Specifically, a pooled HR estimation of 3.51 [2.22; 5.57] for ESR1, 2.80 [2.00; 3.92] for TP53, and 4.56 [2.32; 9.00] for WFDC2 was obtained. Remarkably, WFDC2 has higher HR and 95% CI than ESR1 and TP53, making this protein a good and easy-to-assess biomarker, since it has been identified in serum samples (Figure 8A).

Additionally, the protein expression of these biomarkers was investigated in the data provided by the TCGA and CPTAC high-throughput studies. The same expression pattern is observed for the ESR1 and PGR genes in both studies, showing that the loss of these proteins is related to a worse prognosis. Regarding the TP53 expression, there is an inconsistency between the TCGA and CPTAC studies. Whilst the expression of TP53 is related to low OS in the TCGA study, the contrary is observed in the CPTAC study. TP53 has been widely reported in the literature as a poor EC prognostic factor and has been associated with histological type, histological grade, FIGO stage, myometrial invasion, lymph node status, LVSI, DSS, DFS, and PFS in addition to OS [61,62,63,64]. Thus, the CPTAC study should be interrogated to understand this discrepancy, which could be due to the clinicopathological characteristics of the patients or the limited number of cases included in the deceased group (*n* = 7 vs. *n* = 38 of the living group) [65]. MUC16 and WFDC2 were not tested by RPPA in the TCGA study. However, the expression of these proteins in the CPTAC study was reported and both proteins are likely to be upregulated in deceased patients (Figure 8B,C). These results obtained from the analysis of tissue specimens are in line with the observations obtained in serum samples [50,55,56,57].

## 5. Discussion

This systematic review and metanalysis underline the lack of potential prognostic biomarkers of EC. Among the 2507 articles identified in this review, 398 were deeply analyzed. As a result, this review compiles information of 255 potential biomarkers, which are related to one or more of the clinical prognostic factors in EC, recurrence or survival. Although a large list of biomarkers is described, there are critical issues that hamper their clinical application and that are discussed in this section, from a conceptual, methodological and analytical point of view. Additionally, new strategies in biomarker research are exposed. The main points of this section are summarized in Figure 9.

The REMARK guidelines [27] should be followed to assess the quality of the results derived from biomarker studies. We noted that many articles do not follow these guidelines and important information is missing, limiting the possibilities to reach definitive conclusions of the usefulness of the reported biomarkers.

Study design is of key importance to successfully achieve robust biomarkers. Hence, the investigation of the unresolved problem has to guide the study design. Patients need to be selected according to well-defined inclusion and exclusion criteria, compiling all information possible and in a proper proportion and number to reach the required statistical power. Contrary to this, most of the assessed studies treated the usefulness of prognostic biomarkers as a secondary objective in the study. As a consequence, careful selection of patients is not performed, generating highly heterogenous groups leading to false-positive results. Additionally, statistical power is clearly affected either by a small number of patients per group or unbalanced comparison groups. In order to solve these misrepresentations, some considerations should be taken. In the initial phases (discovery) of the biomarker pipeline the aim will be to have the most balanced and less variable group of patients in order to avoid side effects, while in late phases (clinical) the assessment would be multicentric, prospective and including a wide range of patients covering the heterogeneity of the disease [66].

Another important aspect of the study design is the clinical sample that is selected to perform the study. As shown, most of EC studies were performed using tissue samples (79%), whereas 16% of the studies used serum as a potential source, and only the remaining 5% of the articles used other sources such as imprint smears, peritoneal cytology or uterine aspirates. A convenient and effective biomarker should not only discriminate two groups of patients with high accuracy but if possible, be easily implemented in clinical practice. An ideal prognostic biomarker should be identified preoperatively since this information is relevant to guide the treatment of the patient, which is mainly surgical and can vary from minimal to extensive surgery. To achieve that, a prognostic biomarker in EC should be measurable in noninvasive samples collected at the early steps of the diagnostic process. Therefore, the aim of investigators should be translating the results in easy-to-access biofluids such as blood or proximal biofluids. However, this has not been a priority yet, since only 21% (84/398) of the articles were assessing proteins in non-tissue samples, and remarkably, almost half of them only assessed two proteins, i.e., MUC16 and WFDC2.

EC study design should include the TCGA classification as an additional parameter to either recruit patients or evaluate the results. In this review, we only identified 11 articles including the TCGA classification. The incorporation of the TCGA molecular classification in research and clinical practice for classifying EC patients should be promoted, especially when studying prognostic biomarkers.

The rapid advances in medical and biomedical sciences have a huge impact on the outcome for patients. This is possible thanks to the tight relation between medical identification of clinical needs and the consequent solution from the research side. However, regarding EC disease, even if the clinical gaps are well-known, more research is needed to provide solutions to all of them. Based on our review we identified the lack of discovery studies as one of the main causes. Discovery studies allow for the identification of de novo biomarkers since they screen for the whole or at least, an abundant part of the proteome of the samples that are being studied. Following our search criteria, we could only identify two discovery studies on prognostic factors in EC. Teng et al. unveiled the potential use of PKM2 and HSPA5 as biomarkers of high-risk EC by using two-dimensional gel electrophoresis (2-DE) and a liquid chromatography electrospray ionization tandem mass spectrometry (LC-ESI-MS/MS) proteomics approach [67]. More recently, Yang et al. used reverse-phase protein arrays (RPPA) to study 186 proteins and phosphoproteins in tissue samples in a training (*n* = 183) and validation (*n* = 333) cohort of patients. Their results yielded an algorithm that combined two clinical variables and 18 protein markers to improve risk classification in early-EC [68]. We cannot exclude the possibility that other search criteria might yield additional discovery studies, but probably, they were not labeled as biomarker research studies. Therefore, discovery studies focused on the identification of prognostic biomarkers should be fostered in the near future.

Most of the included studies are focused on the analysis and validation of one or a few proteins. This raises another important issue, which is the lack of biomarker panels. Since cancer is a multifactorial disease, a single biomarker is unlikely to have high accuracy on its own [69]. Related to this, new advances in MS instrumentation, acquisition methods, and associated informatics tools for data processing benefit the scientific community through all phases of the biomarker pipeline, i.e., discovery, verification and validation phases. MS instrumentations are becoming considerably more sensitive and faster, able to achieve higher selectivity and confidence in the peptide identification process [70]. Therefore, it is a great opportunity to screen clinically relevant samples and generate comprehensive lists of candidate biomarkers using different acquisition methods such as data-dependent acquisition (DDA) or the more recently emerged data-independent acquisition (DIA) [71]. Additionally, MS technology can be the perfect platform to further validate these lists of candidate biomarkers in a larger number of patients, since it allows us to highly multiplex and monitor a hundred proteins at once by using methods such as selected reaction monitoring (SRM) or parallel reaction monitoring acquisition (PRM). These techniques have already demonstrated their potential in EC diagnosis either in tissue and biofluids [72]. In our revision, only Martinez-Garcia et al. used a targeted MS approach to assess prognostic biomarkers [54]. In this study, 52 proteins were evaluated in uterine aspirates of a cohort of 116 patients and the results pointed to a 3-protein panel that differentiated endometrioid vs. serous EC histologies with 95% sensitivity and 96% specificity. As seen in this study, combinations of biomarkers provide improved discrimination power over molecular tests based on single markers. Additionally, the integration of molecular biomarkers with clinicopathological features would probably be the most convenient approach to develop more sensitive and specific tests, as we observed in some studies such as in Yang et al. [68].

To achieve clinical implementation of biomarkers is another relevant point of discussion. Once validated, biomarkers should be ideally transferred to a standardized, economic, simple and fast analytical platform and should be prospectively validated following all the regulatory requirements to become an in vitro diagnostic test. Currently, the assessment of a combination of biomarkers would be easy to implement by using a multiplex immunoassay, although another possibility is to perform each biomarker independently through a standard ELISA or immunohistochemistry assay. A new trend to implement multiple biomarkers in the clinical routine is clinical MS, which seems to advance faster in the last years with the commercialization of a new generation of instruments that are fully automated, such as the Thermo Scientific™ Cascadion™ SM Clinical Analyzer (Thermo Fisher).

Finally, a robust statistical analysis of the results is essential to decide whether a protein has sufficient power to become a biomarker. In this systematic review, the vast majority of articles only use the *p*-value to assess the clinical utility of the candidate biomarkers. However, this parameter is not sufficient [73], and it should be supported by other descriptive statistics such as fold-change ratios and parameters used in evaluating clinical validity of a biomarker (e.g., sensitivity, specificity, positive predictive value (PPV), negative predictive value (NPV), AUC value, among others). Additionally, biomarkers should be validated in large and external cohorts of patients. In this respect, an important step in the biomarker field is publicly available data produced by the large consortiums of the CPTAC and the TCGA. These studies include not only proteomic but also genomic and transcriptomic data, all of them associated with clinical features of carefully recruited cohorts [32]. These resources will permit assessing the potential of biomarkers and prioritizing biomarkers in the subsequent phases of the biomarker pipeline. As far as prognostic factors are concerned, applying multiple Cox regression or similar predictive modeling (‘multivariate’ analysis) to adjust for clinical parameters would be the desirable testing. This method would allow us to report the estimated HR and its CI, as suggested by other authors [27]. However, often limited cohort size does not allow to conduct a regression analysis of good predictive performance, which may affect the validity of the estimation results. Indeed, it has been reported that at least 20 individuals per event would be needed for reliable modeling [74]. Moreover, despite in many studies OS being used to predict mortality, DSS should be considered.

## 6. Conclusions

The integration of clinical, molecular and pathological data may improve the current EC risk stratification system, which is crucial to select the most optimal treatment for each patient. Here, we systematically reviewed the literature for EC prognostic biomarkers at the protein level and compiled a list of 255 proteins, although 79% of those proteins would require further validation. The only proteins that have been extensively studied are carbohydrate antigen 125 (CA125 or MUC16), human epididymis protein 4 (HE4 or WFDC2), estrogen receptor (ESR1) and progesterone receptor (PGR), mismatch repair proteins (MMR proteins: MSH2, MSH6, MHL1, PMS2), the tumor suppressors PTEN and TP53, the cell adhesion molecules E-cadherin (CDH1) and neural cell adhesion molecule L1 (L1CAM), the proliferation marker protein Ki-67 (KI67), and the Erb-B2 Receptor Tyrosine Kinase 2 (ERBB2). On the basis of our meta-analysis, ESR1, TP53 and WFDC2 may be useful prognosticators for OS of EC. We also identified critical conceptual, methodological and analytical factors that need to be improved in further research. Consequently, we encourage the scientific community to follow these considerations in order to successfully identify clinically valuable EC prognostic biomarkers: (i) design studies whose primary aim is the identification of prognostic biomarkers. Thus, patient selection should be balanced and controlled to achieve this objective, and the sample of the study source of biomarkers should be carefully chosen; (ii) include high-throughput technologies such as MS to have a broad analysis of biomarkers and to have the feasibility to develop biomarker panels; (iii) the statistical analysis in every step of the biomarker pipeline should be thoughtfully performed.

## Figures and Tables

**Figure 1 jcm-09-01900-f001:**
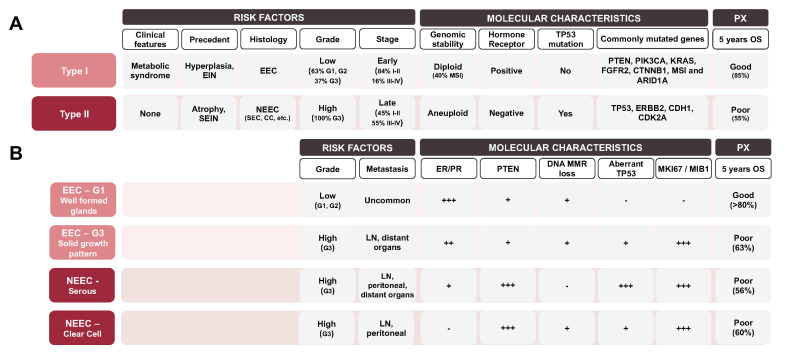
Principle features of the two different subtypes described in the dualistic classification model of endometrial cancer (EC) by *Bokhman* et al. [7] (**A**) Risk factors, molecular characteristics and prognosis of the dualistic classification. (**B**) Deconstruction of the dualistic model according to the different histological grades that exist on endometrioid-endometrial cancers (EECs) and the two most common histological subtypes of non-endometrioid endometrial cancers (NEECs)*. PX: prognosis; OS: overall survival; SEC: serous EC; CC: clear cell EC; EIN: endometrial intraepithelial neoplasia; MSI: microsatellite stability instable; SCNAs: somatic copy number alterations load; Sp.Mlc.alterations: specific molecular alterations; MMR: miss-match repair proteins: SEIN: serous endometrial intra-neoplasia; LN: lymph node status; +++: present/high; ++ frequent; + occasional; - absent/low.*

**Figure 2 jcm-09-01900-f002:**
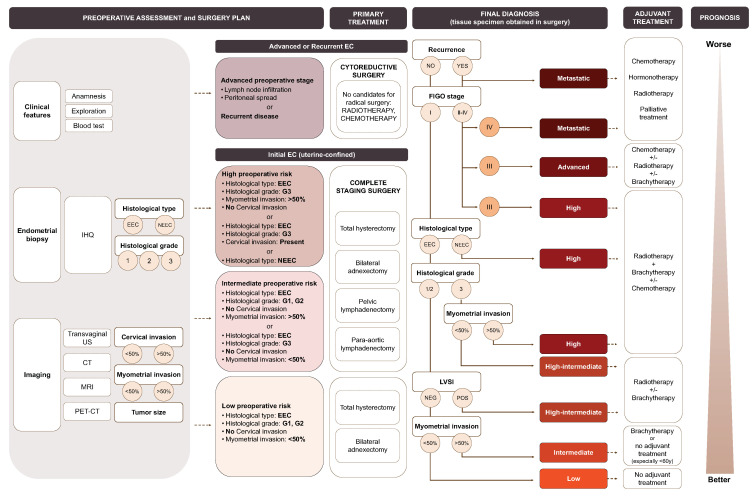
The EC risk stratification system according to the ESMO-ESGO-ESTRO (European Society for Medical Oncology-European Society of Gynaecological Oncology-European SocieTy for radiotherapy & Oncology) consensus [13] and its associated primary and adjuvant treatment. Clinical, molecular and pathological characteristics used to predict EC treatment and decision tree showing the clinical and pathological features used for the final definition of EC treatment. *IHQ: immunohistochemistry; Transvaginal US: transvaginal ultrasonography; CT: computed tomography; MRI: magnetic resonance imaging; PET-CT: positron emission tomography; EEC: Endometrioid endometrial cancer; NEEC: non-endometrioid endometrial cancer; LVSI: lymphovascular space invasion. The information is scaled down to provide a result on the associated risk, primary and adjuvant treatments, and prognosis.*

**Figure 3 jcm-09-01900-f003:**
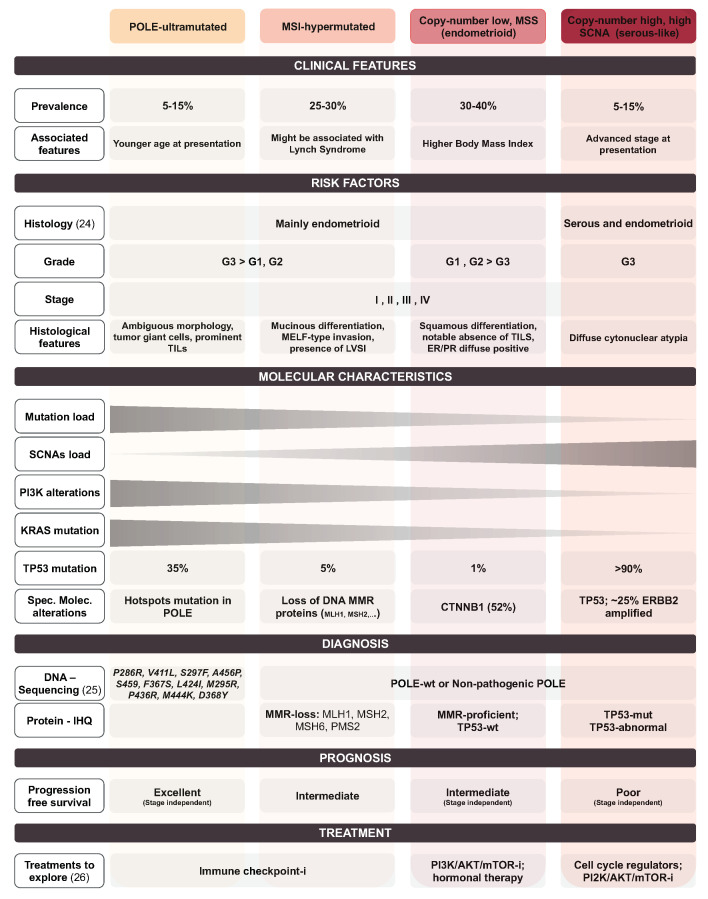
Clinical features, risk factors, molecular characteristics, diagnosis, prognosis and treatment associated with each subgroup of the TCGA classification system [15,16,26]. *MSI: microsatellite stability instable; SCNA: somatic copy number alterations load; IHQ: immunohistochemistry; Sp.Mlc.alterations: specific molecular alterations; MMR: miss-match repair proteins; mut: mutated; wt: wild-type; -i: inhibitors.*

**Figure 4 jcm-09-01900-f004:**
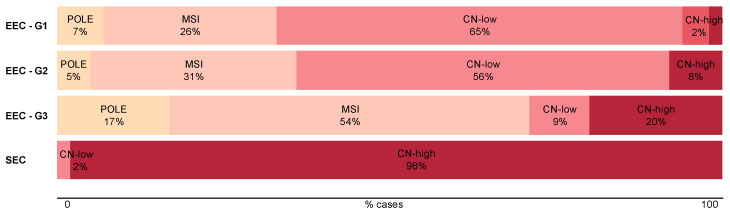
Assessment of TCGA novel classification. Itemization of the TCGA subgroups in the dualistic model. The data used for this figure corresponds to the TCGA cohort [15].

**Figure 5 jcm-09-01900-f005:**
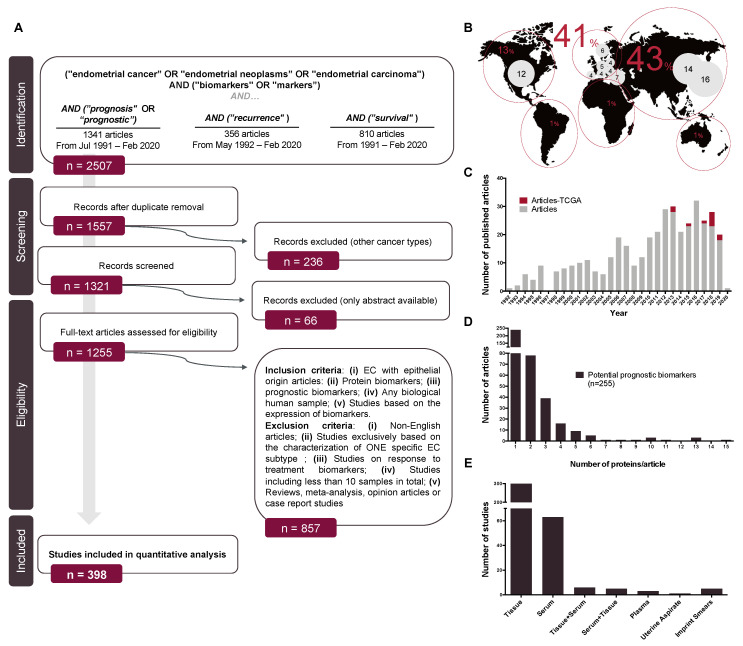
Search strategy and global overview. (**A**) Flow diagram depicting the steps followed for the selection of the studies included in this review; (**B**) world distribution of the selected articles; (**C**) distribution of the selected studies across years. Articles including TCGA classification in their dataset are marked in dark green; (**D**) distribution of the number of protein biomarkers evaluated in each of the studies included in this review; (**E**) Distribution of the studies according to the clinical sample used in the study.

**Figure 6 jcm-09-01900-f006:**
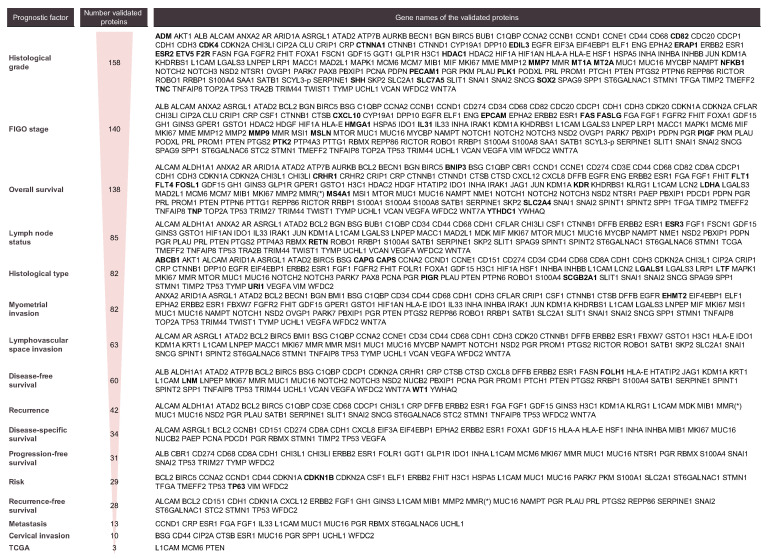
List of proteins associated with each prognostic factor. Proteins linked only to one specific parameter are highlighted in bold.

**Figure 7 jcm-09-01900-f007:**
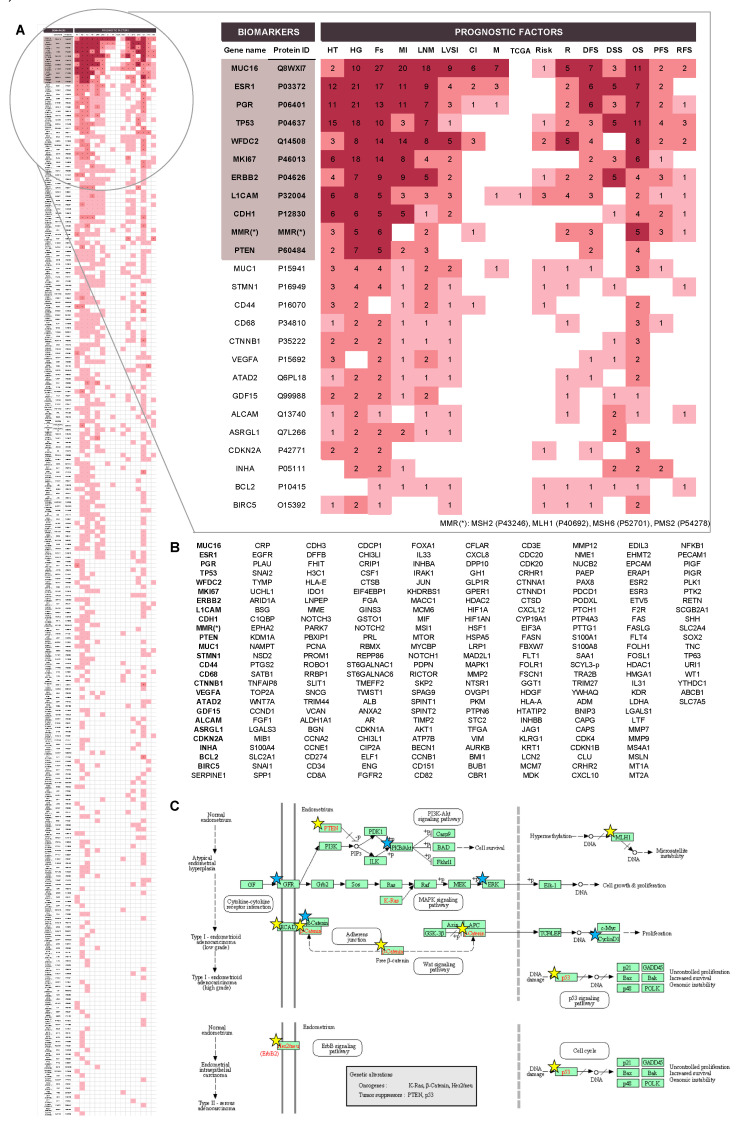
Overview of the validated biomarkers. (**A**) Full perspective of all validated biomarkers: (i) dark red when protein was validated five or more times for that parameter; (ii) red: when protein was validated in more than one study; (iii) light red: protein validated in one study. List of the top-25 most studied proteins as prognostic factor biomarkers is zoomed in. For each protein, the number of studies in which it was validated appears; (**B**) list of proteins validated, at least in one study, for one of the considered parameters. Ordered regarding the number of independent studies where they were validated. In bold, the top-25 most studied proteins; (**C**) EC disease Pathway Map obtained from the KEGG DISEASE database [30,59,60]. The proteins from the 11 most studied proteins list are highlighted by yellow stars, while the top-25 are highlighted by blue stars. *HT: histological type; HG: histological grade; Fs: FIGO stage; MI: myometrial invasion; LNS: lymph node status; LVSI: lymphovascular space invasion; CI: cervical invasion; M: metastasis; TCGA: TCGA classification; R: recurrence; DFS: disease-free survival; DSS: disease-specific survival; OS: overall survival; PFS: progression-free survival; RFS: recurrence-free survival.*

**Figure 8 jcm-09-01900-f008:**
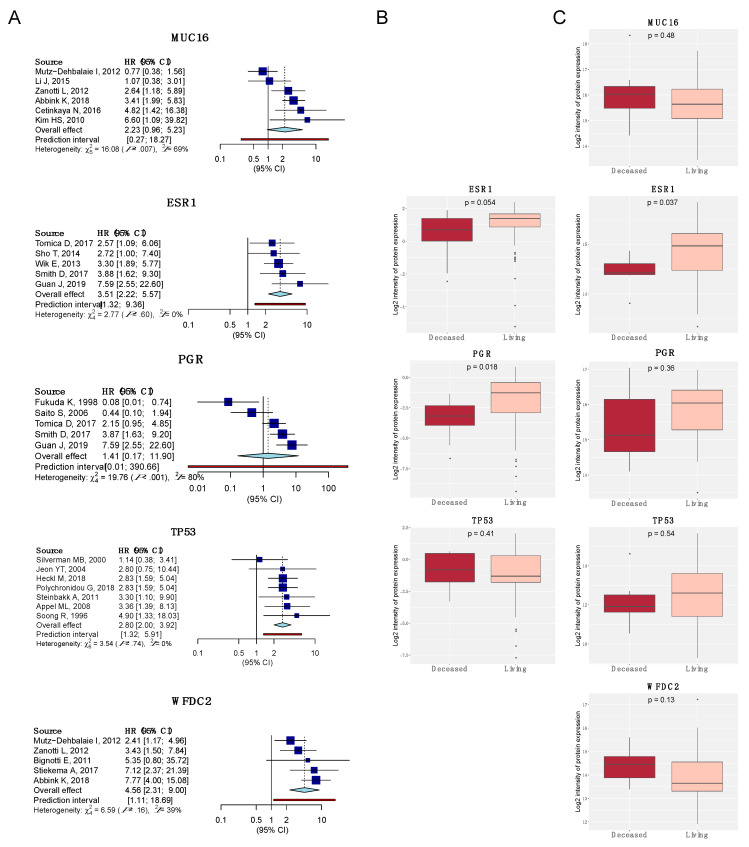
Meta-analysis on OS of the most studied biomarkers regarding prognosis in EC (MUC16, ESR1, PGR, TP53, and WFDC2, respectively). (**A**) Forest plots. Diamond in light blue represents the point estimate and confidence intervals when combining all studies; (**B**) expression boxplots using the RPPA data of the TCGA cohort (*n* = 200:20 deceased–*plotted in red*; 178 living–*plotted in light red*) [15]; (**C**) Expression boxplots using the mass spectrometry data of the CPTAC cohort (*n* = 100:7 deceased–*plotted in red*; 38 living–*plotted in light red*) [32].

**Figure 9 jcm-09-01900-f009:**
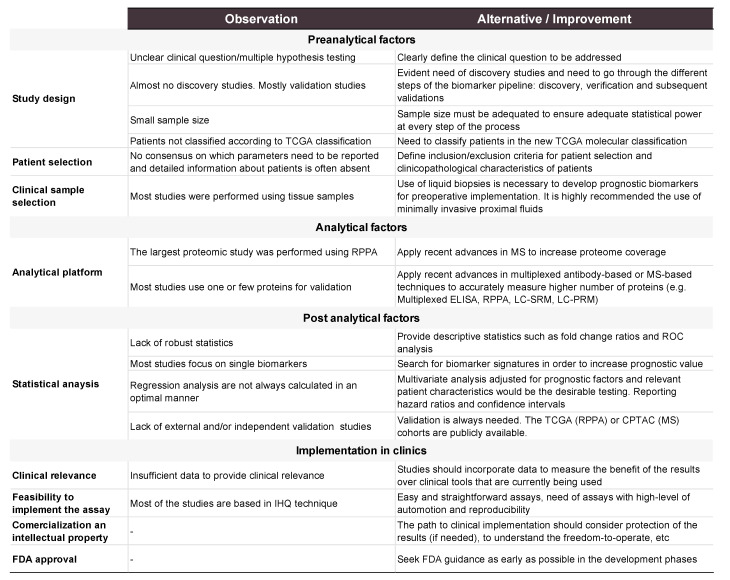
Outline of the preanalytical, analytical, and post-analytical factors detected in the articles reviewed and recommendations of alternatives to consider for future studies.

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
