# Peer review of "Prognostic Biomarkers in Endometrial Cancer: A Systematic Review and Meta-Analysis"

_jcm, 2020, doi:10.3390/jcm9061900_

Round 1
Reviewer 1 Report
The manuscript entitled “Prognostic biomarkers in endometrial cancer: a systematic review and meta-analysis” by Rubia et al. is an interesting article that covers various clinical studies to identify prognostic markers for endometrial cancer. Authors have done a thorough analysis of multiple studies to identify these markers in different clinical samples. In addition, protein markers have been identified instead of expression markers, which may represent relatively closer picture of in vivo scenario. Finally, authors have raised valid points for consideration in future studies.
Author Response
We want to thank reviewer 1 for the evaluation of our manuscript.
Reviewer 2 Report
In this well-conducted systematic review and meta-analysis, Rubia et al included a total number of 398 articles, 255 studied proteins, which have been associated with prognostic factors for endometrial cancer or are directly related to recurrence and survival. Eleven proteins were highlighted as the most extensively validated (in more than 5 independent studies) proteins in the field whereas after meta-analysis of the top-5, they found only 3 with potential usefulness for predicting overall survival in endometrial cancer. Also, limitations of the included studies as well as ways to address these in future studies are described in detail
Comments:
Why have the authors limited their meta-analysis on the top 5 most studied biomarkers? The authors also agree that integration of additional molecular biomarkers is necessary for an improved version of the current stratification system. Why not include all the 11 proteins that were found extensively validated?
Authors should discuss the cost-effectiveness of such an approach with multiple molecular biomarkers which can prove laborious and costly. How do these compare with current approaches?
Minor comments to address
Introduction: Provide worldwide statistics not only US-based
Line 69-70: provide refs
In Fig. 1A, include the percentage of women with low-high grade for the different subtypes rather than “mainly” low and high. Similarly for the Stage. Give abbreviation of MSI in the legend
Line 129: These surrogate biomarkers
Fig 8: correct (c) with (b)
Line 355: These results..are..
